# Draft genome of multiple resistance donor plant *Sinapis alba*: An insight into SSRs, annotations and phylogenetics

**Preetesh Kumari[1‡], Kaushal Pratap Singh [2‡]\*, Pramod Kumar Rai[2]**

**1** ICAR-National Institute on Plant Biotechnology, IARI, Pusa Campus, New Delhi, India, **2** ICAR-Directorate of Rapeseed Mustard Research, Sewar, Bharatpur, Rajasthan, India

‡ These authors share first authorship on this work.
\* kaushalmpi1978@gmail.com

## Abstract

### Background

*Sinapis alba* is a wild member of the Brassicaceae family reported to possess genetic resistance against major biotic and abiotic stresses of oilseed brassicas. However, the resistance nature of *S. alba* was not exploited generously due to the unavailability of usable genome sequences in public databases. Therefore, the present study was conducted to assemble the first draft genome from raw whole genome shotgun sequences with annotation and develop simple sequence repeat markers for molecular genetics and marker-assisted breeding.

### Results

The raw genome sequences had 96x coverage on the Illumina platform with 170 Gbp data. The developed assembly by SOAPdenovo2 has ~459 Mbp genome size covered in 403,423 contigs with an average size of 1138.04 bp. The assembly was BLASTX with *Arabidopsis thaliana* which showed 32.9% positive hits between both plants. The top hit species distribution analysis showed the highest similarity with *A. thaliana*. A total of 809,597 GO level annotations were recorded after BLASTX results, and 34,012 sequences were annotated with different enzyme codes grouped under seven classes. The gene prediction tool AUGUSTUS identified 113,107 probable genes with an average size of 684 bp. The biochemical pathway annotation assigned 16,119 potential genes to 152 KEGG maps and 1751 enzyme codes. The development of potential SSRs from the *de-novo* assembly yielded 70731 unique primer pairs. Out of 159 randomly selected SSR markers for validation, 149 successfully amplified in *S. alba*. However, 10 SSR markers did not amplify during the validation experiment.

### Conclusion

The annotated genome assembly with a large number of SSRs was developed in the present study. To the best of our knowledge, this is the first report of *S. alba* genome assembly

---

**Data Availability Statement:** All relevant data are within the paper and its supporting information files. This Whole Genome Shotgun project has been deposited at DDBJ/ENA/GenBank under the

accession WIDR00000000. The version described in this paper is version WIDR01000000.

**Funding:** 1. Preetesh Kumari, File No. YSS/2015/001849), Science and Engineering Research Board, Ministry of Science & Technology, India, https://www.serbonline.in, Sponsor does not have any role in the study design, data collection and analysis, decision to publish, or preparation of the manuscript 2. Kaushal Pratap Singh, File No. 19/1247(0001)/19-EMR-I, HRDG-Council of Scientific & Industrial Research, New Delhi, www.csirhrdg.res.in, Sponsor does not have any role in the study design, data collection and analysis, decision to publish, or preparation of the manuscript.

**Competing interests:** The authors have declared that no competing interests exist.

development, annotation, and SSRs mining to date. The data presented here will be a very important resource for future crop improvement programs, especially for resistant breeding.

## 1. Introduction

Next-generation sequencing (NGS) provides high throughput data for analyzing complete genomes of diploid and polyploidy plants. In recent years, NGS is being used worldwide for generating sequence information that can be used for decoding numerous eukaryotic genomes. In agriculturally important crops including their wild donor plants, the reference or draft level genome sequences and its annotation information can be used for the rapid identification of potential genes. These sequences can also be used to develop molecular markers in the form of single nucleotide polymorphism and simple sequence repeats (SSRs). These markers are used to construct fine linkage maps and map-based cloning of important genes and are also used in marker-assisted breeding to accumulate rapid information about plants with useful traits. The identified genes can be marked for their roles in different biological pathways of plant systems. These marked genes can be cloned and introgressed in economically useful crop plants to develop super agriculturally-important crops. In the Brassicaceae family, *Arabidopsis thaliana* is the completely sequenced model plant that is used as a reference genome to annotate unknown gene sequences.

The Brassicaceae family contains 338 genera and 3,709 species including oilseed crops, vegetables, and some wild relatives [1]. The white mustard (*Sinapis alba* L., 2x = 2n = 24), a diploid wild member of this family, is widely cultivated for food condiments in dry temperate and tropical countries (e.g., China, India, Pakistan, Russia, Tajikistan, Turkmenistan, Uzbekistan, Vietnam, Africa, Europe, Canada, and North America). It also performs well in dry and hot environments. The plant possesses many important agronomic traits such as tolerance/resistance against *Alternaria* blight [2,3,4], *Sclerotinia* stem rot [5], beet cyst nematode [6], flea beetles [7,8], pod shattering [9,10], high temperature [11], and drought [12]. Currently, the importance of this plant has increased due to the identification of antioxidants, antibiotics, and anti-cancer chemicals in its seed extract [13]. The plant can absorb heavy metal (Cd) from the soil, suggesting its potential application in phytoremediation [14]. The cultivated crop brassicas have very low genetic variability and are highly susceptible to many diseases, resulting in heavy yield losses [15,16,17,18]. The *S. alba* has post-fertilization barriers with *Brassica juncea*; thus, resistance could not be transferred. However, some workers developed somatic hybrids of *S. alba* with cultivated oilseed brassicas to introgress resistance [4,19]. The *S. alba* genome assembly was not available till now, so all useful work like molecular marker discovery and identification of genes could not be possible previously.

Compared to other Brassicaceae family plants such as *Brassica rapa*, *Brassica nigra*, *Brassica oleracea*, *B. juncea*, *Brassica napus*, the genome sequencing data of *S. alba* was not available in the public domain. However, the report on the transcriptome assembly is available and published [20]. Thus in this study, we have used Illumina sequenced NGS data of *S. alba* genome from NCBI (Accession No. SRR3490913, SRR3490914, SRR1799170, SRR1799171). This raw data was used to develop *de-novo* assembly, gene prediction with annotations and identification of SSR markers. The insight of *S. alba* genome provides useful information on SSRs, genes and biochemical pathways that could be used in crop improvement programs.

## 2. Material and methods

### 2.1. Assembly preparation

Department of Energy, Joint genome institute, USA had developed whole genome shotgun (WGS) sequences (Illumina Hiseq 2500/2000) of *S. alba* (Strain: S2 GC0560-79) and submitted to SRA (NCBI). A total of four different runs were developed by JGI and submitted with accession numbers SRR3490913 and SRR3490914 (Single-end) and SRR1799170 and SRR1799171 (Paired-end). We have downloaded WGS sequences of all four runs from NCBI in FASTQ format to develop the first draft genome assembly. The raw sequences were checked for adapter contamination and poor quality bases by FASTQC v0.11.8 software [21]. The identified adapters and poor-quality bases were removed with CutAdapt v2.3 [22] with the quality score (Q) 25 and rechecked with FASTQC. The statistics of all four runs were cross-checked by Assembly-stats software (https://github.com/sanger-pathogens/assembly-stats) before and after the removal of adapter and poor-quality bases. The trimmed quality raw sequences were then used for assembly preparation by SOAPdenovo2 (https://github.com/aquaskyline/SOAPdenovo2) with parameter K = 37 and -R option enabled to solve the tiny repeat region assembly using the reads.

### 2.2. SSR detection and primer design

The potential microsatellite markers were identified in 403,423 contigs by using the MIcroSAtellite (MISA) identification tool (http://pgrc.ipk-gatersleben.de/misa/misa.html). The parameters were adjusted for identification of perfect di-, tri-, tetra-, penta- and hexanucleotide motifs with the minimum number of 6, 5, 5, 5, and 5 repetitions, respectively. However, mononucleotide repeats were not considered for this purpose. The primer pairs were designed by using Primer3 v0.4.0. The parameters used for primer pair designing were put as follows: primer size 18–25 bp (optimum 20 bp), PCR product size 100–400 bp (optimum 280 bp), GC percent 45–70 (optimum 50%), Tm 57–63˚C (optimum 60˚C). Based on the above parameters, 78067 primer pairs were designed.

### 2.3. SSR marker validation

For validation of randomly selected 159 SSR markers, the young leaves of field-grown single *S. alba* (Strain: DRMR-2183) plant were used for genomic DNA extraction and purification as reported earlier [23]. Nanodrop 8000 spectrophotometer was used for quantification (Thermo Fisher Scientific, USA). The 20 μl PCR reaction mixture contained template DNA- 2 μl (25 ng/μl), primer pairs (forward and reverse)- 2 μl (10 ng/μl), 0.05 mM dNTP mix- 0.25 μl, 2.5 mM MgCl2−0.25 μl, Taq DNA polymerase buffer (10 X)- 2 μl, 0.25 U Taq DNA polymerase (Banglore GENEI)- 0.5 μl, and nuclease-free water- 13 μl. The PCR conditions were 4 min at 94˚C, followed by 40 cycles at 94˚C/30 sec (denaturation), 58˚C/30 seconds (primer annealing) and 72˚C/30 seconds (primer extension), followed by a final extension at 72˚C for 7 min. To determine the approximate amplicon size, 50 bp DNA ladder (New England Biolabs) was used as a molecular marker. PCR products were resolved on a 3% agarose gel and visualized by ethidium bromide staining in a gel documentation unit.

### 2.4. Structural and functional annotation of assembly

Before downstream analysis of *de-novo* assembly, repeat masking was done with Repeat Masker. A local BLASTX was used to compare the similarity between *S. alba* and *A. thaliana* by the OmicsBox tool (https://www.biobam.com/omicsbox/). The protein fasta (downloaded from NCBI; RefSeq assembly accession No. GCF_000001735.4_TAIR10.1) of this plant was

used for local BLASTX at an *E* value threshold of 1e-25 with all unique contigs (4,03,423) of *S. alba*. The minimum alignment length (hsp-length) was set on 11 bp. The BLASTX result was used in OmicsBox for Gene Ontology (GO) analysis. OmicsBox retrieves the most significant GO terms related to the blast hits of query contigs. The retrieved GO terms were owed to query contigs and classified into cellular component, molecular function, and biological process categories. The contigs were also used for gene prediction by AUGUSTUS webserver (*ab-initio*) through OmicsBox. The predicted genes were further searched for similarity by NCBI blast+ via CloudBlast on BLASTX program with UniProtKB/Swiss-Prot (swissprot_v5) and Non-redundant protein sequences (Nr_v5) databases. The taxonomic groups were selected as Brassicaceae and Brassica to specify our search results. The filter was used as a BLAST against a subset of taxonomies. The genes were further used to annotate against the Kyoto Encyclopedia of Genes and Genomes (KEGG) metabolic pathways database [24]. The pathway genes related to plant physical defense systems were identified and verified by BLASTN with available information against the NCBI database. The key gene in plant cuticular wax synthesis was identified and examined their genetic relationship within the Brassicaceae family. The MEGA software was used to construct a dendrogram based on the unweighted pair group method of an arithmetic average algorithm from gene sequences present in various plant species [25]. The *S. alba* genome was examined for all types of retrotransposons and classified according to their category.

## 3. Results and discussion

*S. alba* could be a potential donor of genetic resistance for biotic disease, environmental stresses, and some other agronomic traits to close relatives such as *B. juncea*, *B. napus* and other brassica crops [12,26,27]. Much research has been focused on this wild plant for its medicinal values because of the presence of anti-bacterial, antioxidant, and anti-cancer chemicals [13]. The processed genome sequencing data was not available in any database. However, the transcriptome sequencing was reported for glucosinolate and phytochelatin pathways [20]. The raw sequenced data of transcriptome and whole nuclear genome have been submitted in NCBI. In the present study, WGS sequence data was used for assembly, annotations, and SSRs development.

### 3.1. *De-novo* assembly development

The whole genome shotgun sequences of *S. alba*, developed and submitted to NCBI by the Department of Energy, Joint genome institute, USA was used for *de-novo* assembly development. The sequences were submitted in four accessions generated from four different runs. The total genome coverage in these runs was 96x on Illumina Hiseq 2500/2000 platform. The generated FASTQ sequences were single- and paired-end reads. The total raw sequences were approximately 170 G base pairs (Table 1).

At first, the downloaded raw sequences were checked by Assembly-stats software and data statistics were matched with the information provided by the submitter. The sequence length of the raw read was 150 bp, and the total GC percentage was 37. The low-quality bases were trimmed at Phred score (Q) 25, and all identified adapters were removed with CutAdapt 2.3 software. The trimmed raw sequences were analyzed with Assembly-stats and FASTQC for quality check. The trimmed quality raw sequences were then used for *de-novo* assembly preparation using SOAPdenovo2 software with optimum k-mer size 37. A total of 4,03,423 contigs with sum base pair of 459,115,215 were generated as an output of the *de-novo* assembly. The generated draft assembly (459 Mb) was approximately 83% of the total genome size of *S. alba* (553 Mb) [28]. The maximum and minimum contig size was 29,081 and 200 base pairs

**Table 1. Raw Illumina sequenced data statistics.**

| S. No. | Accession No. | Library layout | Read count | Base count |
|---|---|---|---|---|
| 1 | SRR3490913 | Single end | 196,510,340 | 49,324,095,340 |
| 2 | SRR3490914 | Single end | 212,528,446 | 53,344,639,946 |
| 3 | SRR1799170 | Paired end | 150,756,709 | 45,227,012,700 |
| 4 | SRR1799171 | Paired end | 073,922,772 | 22,176,831,600 |

respectively. However, the average size of contigs was 1138.04 bp and N50 was 1742 bp. The contigs below 1000 bp were recorded at 68.19%, and only 25 (0.006%) contigs were found above 20,000 bp (Fig 3A).

## 3.2. SSRs development, characterization and validation

Molecular markers are a vital source of genetic studies, but they are still rare in the case of *S. alba* because of the unavailability of the reference genome in the public domain. The SSRs and intron length polymorphism markers were developed and used by some workers for such studies [20,29,30]. In this study, the generated draft assembly of *S. alba* was used for *in-silico* SSR prediction and flanking primer designing with MISA and Primer3 software [31]. The total number of SSR loci identified in the assembly was 109,980, while the primer pairs were designed only for 78,067 (70.98%) SSR loci. No primer pairs were designed for 31,913 (29.02%) SSR loci. An average of one SSR per 5.88kb genome sequence was developed in this study; similar observations were reported by previous authors [32,33]. However, the average size (kbp) may be varied depending upon the size of datasets (genes, contigs, and scaffolds), program or parameters [34]. The redundant primers were removed, and a total of 70,731 (64.31%) unique SSR markers of *S. alba* were obtained (S1 Table). The dinucleotide motifs were most common SSRs, accounting for about 85.06%, followed by tri-, tetra-, penta-, and hexanucleotide repeats that accounted for 12.99%, 0.94%, 0.35% and 0.46% SSRs, respectively [35]. The developed SSRs also contained 178 (0.16%) hepta-, 8 (0.007%) oct-, 27 (0.024%) nona- and 1 (0.0009%) decanucleotide motifs. Most of the contigs (32261) have 5 repetitions of motifs within SSR length while only one contig has 99 repetitions. Only 24 contigs were found with more than 50 repetitions. (Fig 1A; S2 Table). The top dinucleotide and

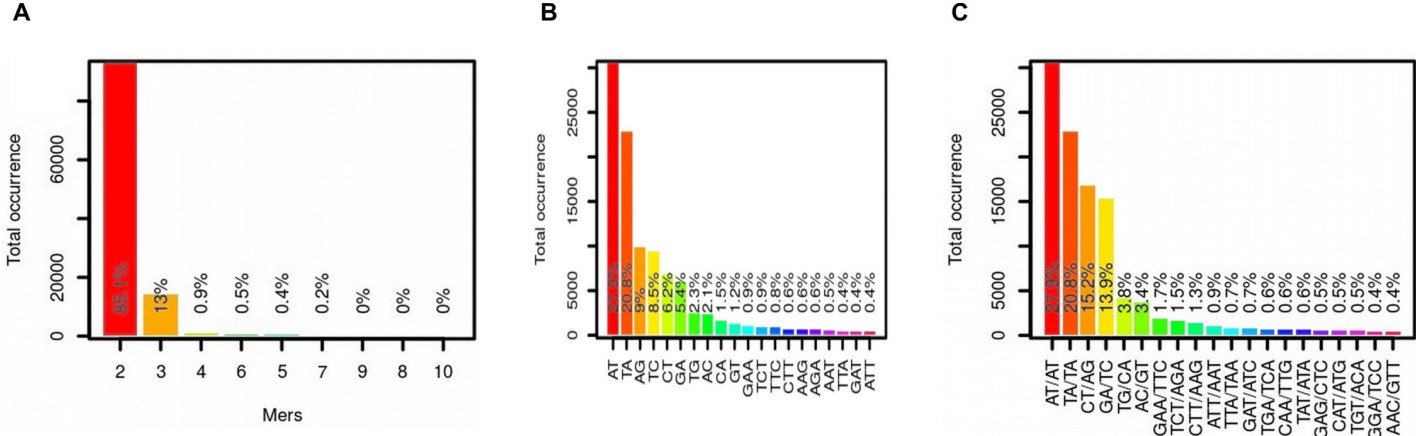

**Fig 1. SSRs development statistics.** (A) Motif distribution against the total occurrence. (B) Top motif distribution in SSRs (C) Abundance of various repeat motifs in total SSRs.

trinucleotide motifs present in the SSRs were AT (27.9%), TA (20.8%), AG (9%), TC (8.5%), CT (6.2%), GA (5.4%), TG (2.3%), AC (2.1%), CA (1.5%), GT (1.2%), GAA (0.9%), TCT (0.9%) and TTC (0.8%) (Fig 1B). This data propensity was also reported in other plant genomes [36]. The frequency of top grouped motifs in SSRs was AT/AT (27.9%), TA/TA (20.8%), CT/AG (15.2%), GA/TC (13.9%), TG/CA (3.8%), AC/GT (3.4%), GAA/TTC (1.7%), TCT/AGA (1.5%), and CTT/AAG (1.3%) (Fig 1C). The detailed information on SSRs included for each marker were SSRs ID, the position of SSRs flanking region in the contig, length and sequence of forward and reverse primers, number of alleles, melting temperature and expected PCR product size. In an earlier study, the transcriptome assembly of 46 days old *S. alba* (ZYZ-1553) plant was developed and produced 14,727 simple sequence repeats from unigenes longer than 1 kb size [20]. However, this plant was attacked by pathogens and environmental stresses during and after the flowering stages. The AUGUSTUS predicted a total of 113107 genes in our study, out of which more than 91800 genes were smaller than 1kb size. There may be a situation that potential gene products were not covered in transcriptome assembly. In the present study, simple sequence repeat markers were developed from both exons and introns in ~5x numbers than transcriptome assembly. Thus, the results of the present study would be more useful over previously reported transcriptome assembly and markers especially when resistance breeding objective included. In a single experiment of 159 randomly selected SSR markers from different repeat motifs, 149 were successfully amplified in *S. alba*. However, 10 markers did not give amplification in *S. alba* (S3 Table). A total of 144 markers produced expected amplicon size, while 5 markers gave single amplicons of larger than expected size. The PCR amplification of only 38 markers is shown in Fig 2. However, two markers (04 and

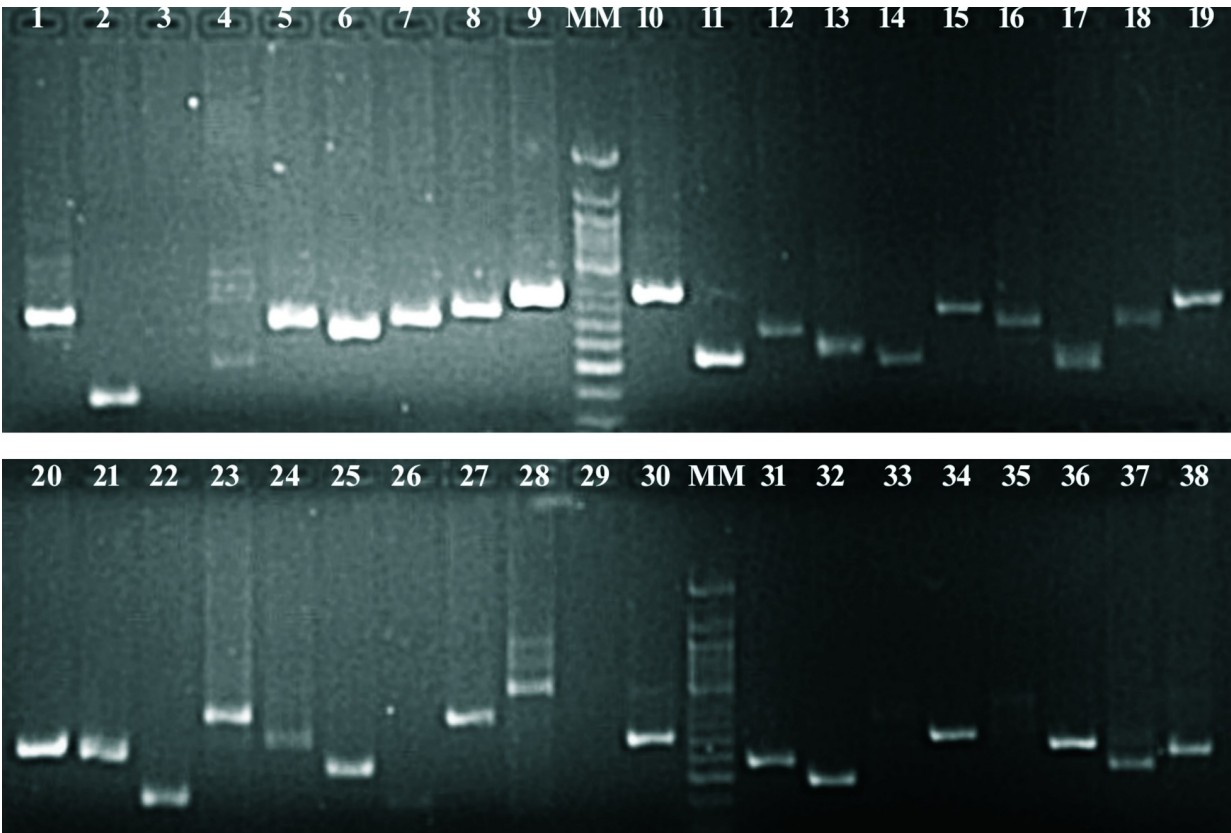

**Fig 2. PCR amplification patterns of tested SSR primer in *S. alba* DNA (Details of primer pairs are included in S3 Table).**

28) were developed more than one amplicon on amplification. Some previous workers reported 65–70% amplification in SSR marker validation experiments [37,38]. However, the flex has been reported at about 92% SSR amplification rate, which was very close to the results of our study [39].

This experiment was the first report of *S. alba* genome assembly and SSRs development. The huge quantity of SSRs developed in this study will serve for crop improvement programs. They will also provide opportunities to work on resistance breeding, marker-assisted breeding, linkage mapping, and transgenic development.

### 3.3. Structural and functional annotation of assembly

The OmicsBox tool was used for all downstream analysis of assembled *S. alba* genome. Before starting genome characterization, the contigs of *S. alba* were subjected to repeat masking by RepeatMasker software using NHMMER search engine. The results showed that a total of 109,639,44 bp (2.39%) were masked; of them, 1.72% belong to simple repeats, 0.67% belong to low complexity DNA (0.67%) (S4 Table).

The blast search revealed a total of 132,763 (32.9%) unique matches between *S. alba* and *A. thaliana*, and the remaining 270,660 (67.1%) contigs did not show any hit (Fig 3B). The contigs were matched at the *E*-value threshold between 1e-4 to 1e-180 (Fig 3C). The similarity distribution of BLASTX hits was recorded from 28 to 100%, while the maximum number of hits (28,580) was found at 60%. A total of 19476 hits showed an absolute similarity between sequences of both plants. A total of 25.44% hits showed higher than 80% similarity; while hits with the similarity between 28 to 80 percent were 74.56% (Fig 3D). The smallest contig (220 bp) showed 9% annotations, and the largest contig (29081 bp) showed 100% annotations (Fig 3E). The InterProScan results showed that a total of 403,277 contigs were analyzed, out of which 265,913 were without IPS, 137,510 with IPS and 52,741 with GOs terms. The Tag distribution analysis revealed that 108,748 and 125,438 contigs were annotated and mapped, respectively (Fig 3F).

The homologous sequences of SOAPdenovo2 assembled contigs were assigned to GO terms. The highest number of GO terms (1124) was recorded in contigs with the length of 749 base pairs, while many contigs showed only 1 GO term. A total of 809,597 annotations were found based on BLASTX results (Fig 3G). A total of 34,012 sequences were annotated and grouped under seven enzyme code classes, i.e., Oxidoreductases (14.23%), Transferases (35.7%), Hydrolases (36.15%), Layases (5.72%), Isomerases (2.58%), Ligases (2.55%) and Translocases (3.07%) (Fig 3H). The GO database is a useful source as the GO terms assigned to genes of any organism provide dynamic, controlled vocabulary information about its role [40]. The *S. alba* contigs were divided into three classifications at level 3 based on their functions by BLASTX analysis, i.e., biological process, cellular components, and molecular function. A total of 13,18,309 GO terms were associated with all these contigs. Out of these, the majority were assigned to cellular components (483,328; 36.66%), followed by biological process (607,567; 46.09%) and molecular functions (227,414; 17.25%) (Fig 4). The contigs involved in biological process were shown to have putative function in organic substance metabolic process (9%), cellular metabolic process (8%), primary metabolic process (8%), nitrogen compound metabolic process (7%), biosynthetic process (4%), response to stress (4%) and chemical (4%), response to abiotic (3%), and so on. Some other components such as cell communication, response to biotic stimulus, signal transduction regulation of biological quality or catabolic process were each represented by ≤ 2% of the total (Fig 4A). The contigs were classified in the cellular component category; the majority was found in terms of intracellular (17%), intracellular part (17%), intracellular organelle (15%), membrane-bound organelle (15%), cell periphery (6%), etc. (Fig 4B). The contigs classified in the molecular function category were shown to

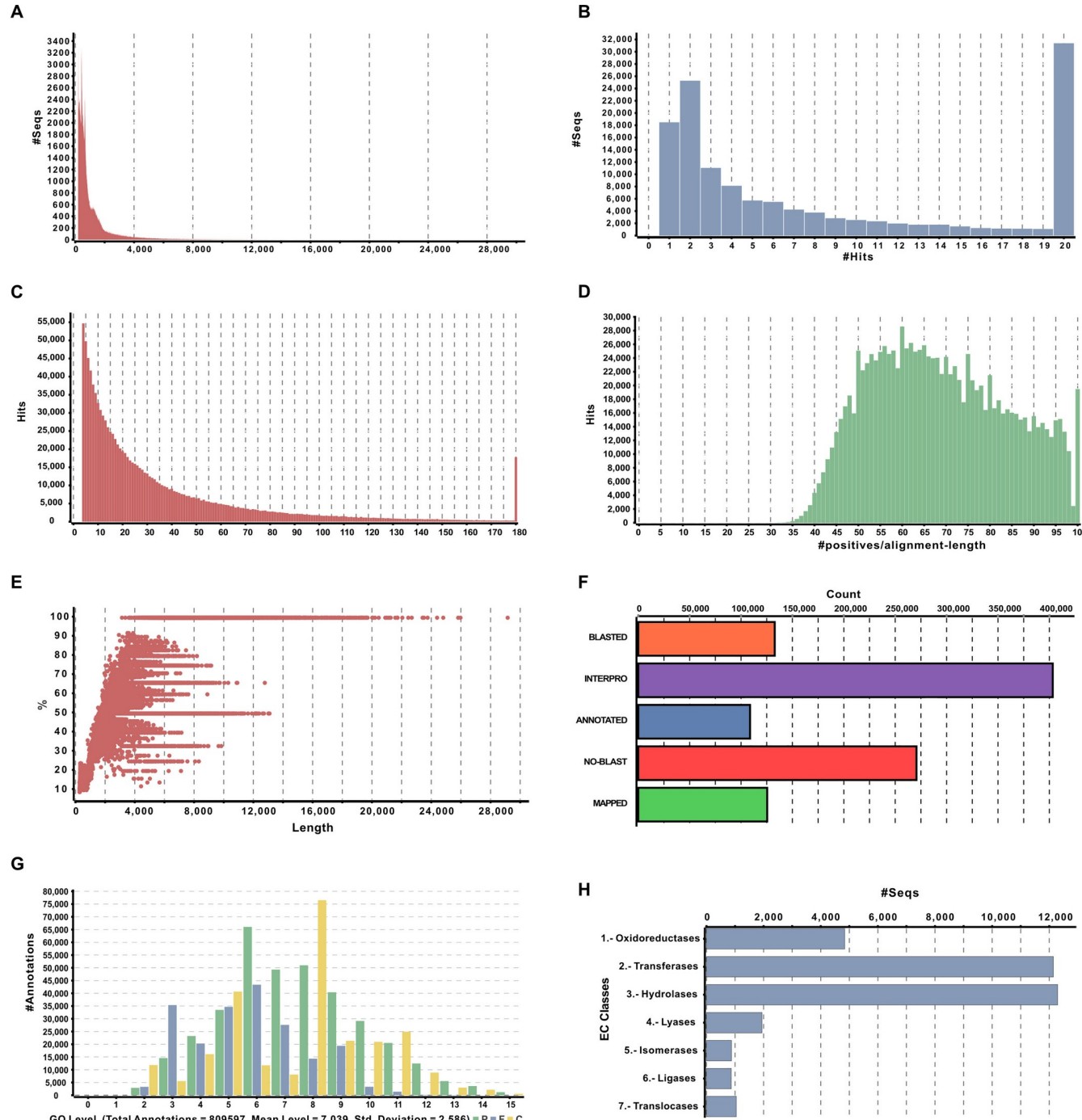

**Fig 3. Structural and functional annotation of assembly.** (A) Sequence length distribution of *de-novo* assembly. (B) BLASTX hit distribution. (C) *E*-value distribution of local BLASTX hits for each contig. (D) Similarity distribution of BLASTX hits for each contig. (E) Annotation percentage of all contigs against length. (F) Four-step distribution of OmicsBox process including BLASTX, IPScan, Mapping, and Annotation. (G) GO level distribution for Biological process [P-BP, F-MF, C-CC], Molecular function and Cellular component. (H) Enzyme code distribution for major classes.

have putative function in organic cyclic compound binding (17%), heterocyclic compound binding (16%), protein binding (13%), transferase activity (10%), catalytic activity (6%), drug binding (5%), and so on (Fig 4C).

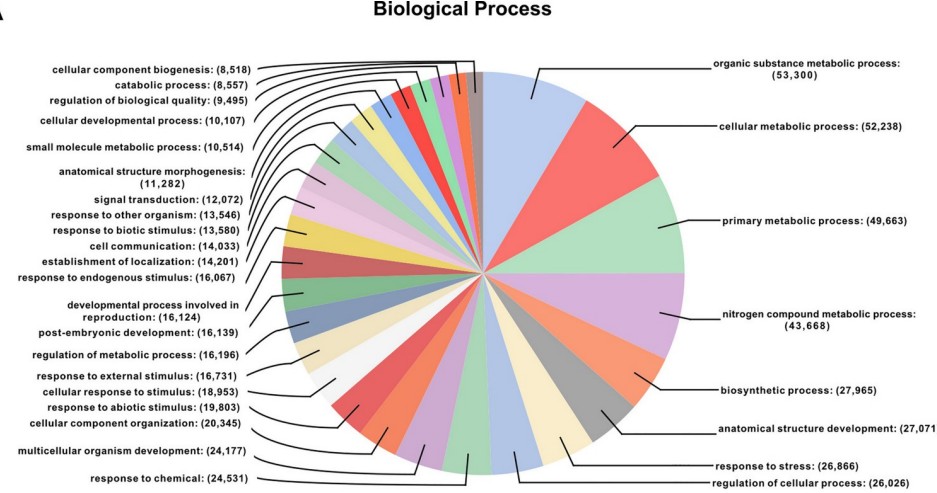

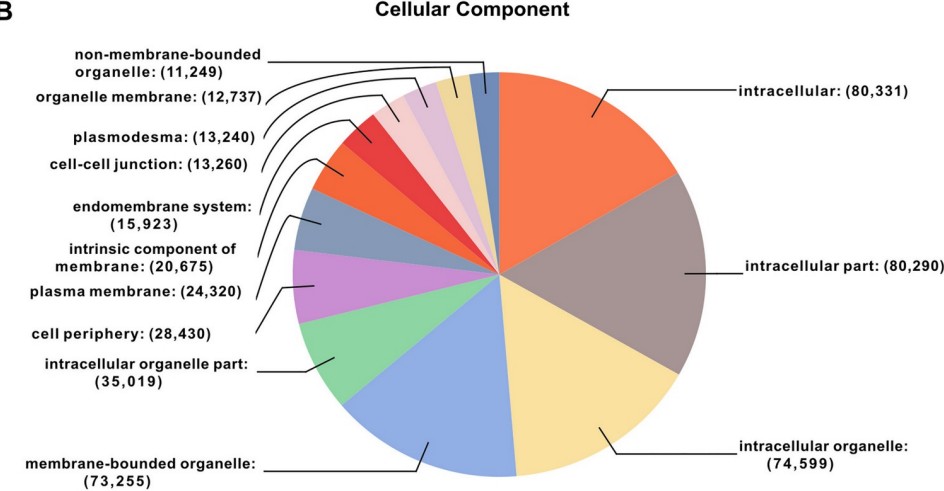

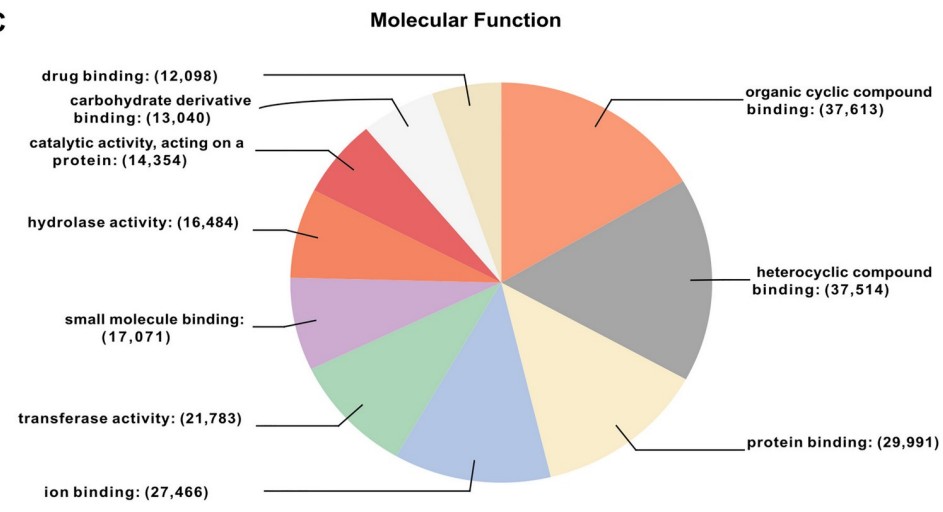

**Fig 4. Pie-chart representations for classification of *S. alba* assembly into functional categories according to GO terms.** (A) Biological process, (B) Cellular component, (C) Molecular function.

## 3.4. Gene prediction and annotation

All *S. alba* contigs were used for gene prediction by AUGUSTUS webserver (*ab-initio*) [41,42] through OmicsBox. The results revealed a total of 113,107 genes with an average length of 684 base pairs. The predicted genes covered a total of 77,411,493 base pairs with minimum and maximum gene length of 102 and 10359 base pairs, respectively. A total of 81.2% predicted mRNA sequences were found below 1000 base pairs (Fig 5A). The coding potential distribution showed that 98% (110,690) predicted genes have the protein-coding ability, while only 2% (2417) genes may not produce a functional product. The coding probability check showed that 96,327 (85.16%) predicted genes had more than 90% coding potential. The analysis progress report of predicted genes revealed that 113,107 sequences have InterProScan IDs. The locally BLASTX search with *A. thaliana* sequences for similarity made hits in 78,625 sequences. The total numbers of sequences mapped and GO annotated were 77,698 and 56,160, respectively (Fig 5B).

The predicted genes with near-related species were compared with UniProtKB/Swiss-Prot (swissprot_v5) and Non-redundant protein sequences (Nr_v5) databases. The species distribution analysis discovered that the predicted genes of *S. alba* had homology with several other plant species. The top hit species distribution showed the highest similarity of *S. alba* with *A. thaliana* (42,299), followed by *B. napus* (291), *S. alba* (74), *B. juncea* (45), etc. (Fig 6A). The species distribution also revealed maximum similarity with *A. thaliana* (128,494), *B. napus* (944), *S. alba* (221), *B. oleracea* (162), *B. juncea* (159) and *B. rapa* (122) (Fig 6B). The Nr database also gave similar results as SwissProt.

The gene function analysis against the KEGG database resulted in assigning a total of 16,119 predicted genes to various biochemical pathways. These genes were found related to 152 KEGG maps and produced 1751 enzyme codes. In this study, there was more than one gene involved in single enzyme annotation against the KEGG database, and similar findings were reported by other studies [43,38]. The KEGG database-annotated genes were categorized under five major pathway groups: metabolism (14,248), genetic information processing (25), environmental information processing (88), organismal systems (1756), and human disease (2). The highest representation was recorded from the metabolism pathways group. The top five pathways within this group were the metabolism of cofactors and vitamins (2814), nucleotide metabolism (2474), xenobiotics biodegradation and metabolism (2014), amino acid

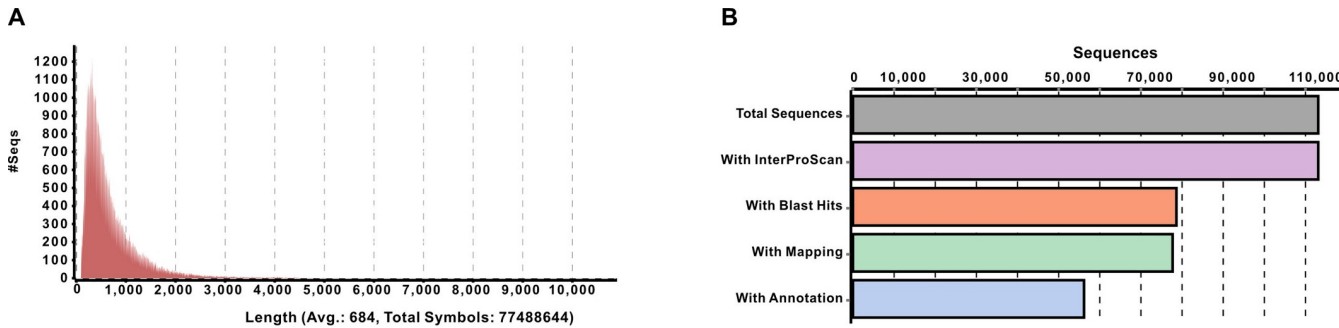

**Fig 5. Sequence length and four-step distribution of *S. alba* genes.** (A) Sequence length distribution of predicted genes by AUGUSTUS. (B) Four-step distribution of genes by OmicsBox including BLASTX, IPScan, Mapping, and Annotation.

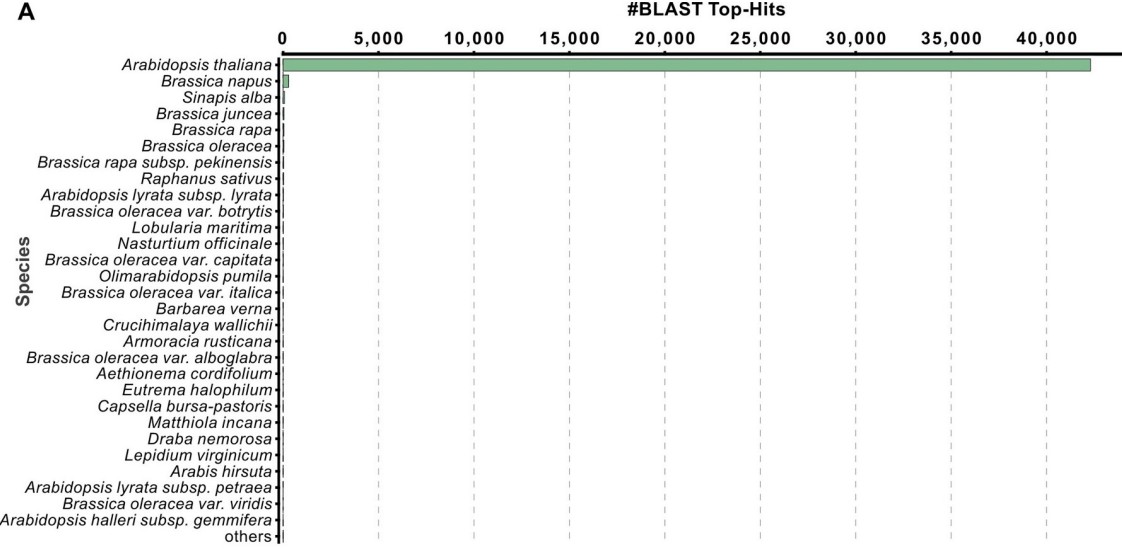

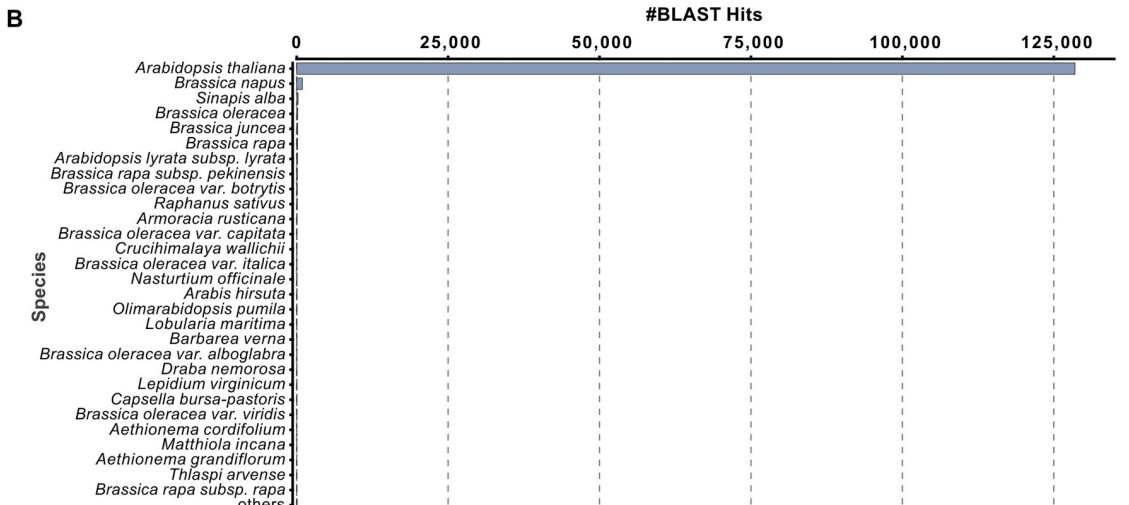

**Fig 6. Species distribution pattern of *S. alba* genes.** (A) Top BLASTX alignment hit species distribution. (B) All BLASTX hit distribution.

metabolism (1472) and carbohydrate metabolism (1467). However, the second-highest group (organismal systems) has three important pathways under the immune system category which are related to the anti-cancer pathways of humans. These three pathways were T cell receptor signaling pathway, PD-L1 expression and PD-1 checkpoint pathway in cancer, Th1 and Th2 cell differentiation with 586, 585 and 585 predicted gene sequences, respectively (S5 Table) (Fig 7).

### 3.5. Reconstruction of defense related pathway genes

*S. alba* is known for its high resistance responses against biotic and abiotic stresses. The outer plant surface is covered by thick cuticular wax and prominent trichomes that protect the plant from drought, pests, pathogens and other external stresses [36,8]. The outer plant surface composed of three major components i.e. cutin, suberin, and wax. A total of ten genes were

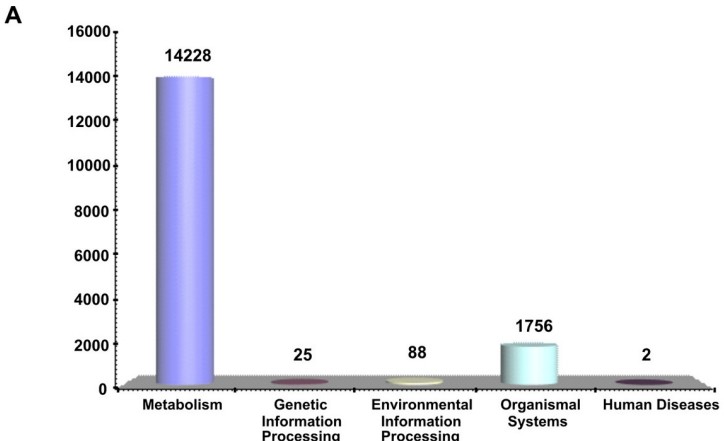

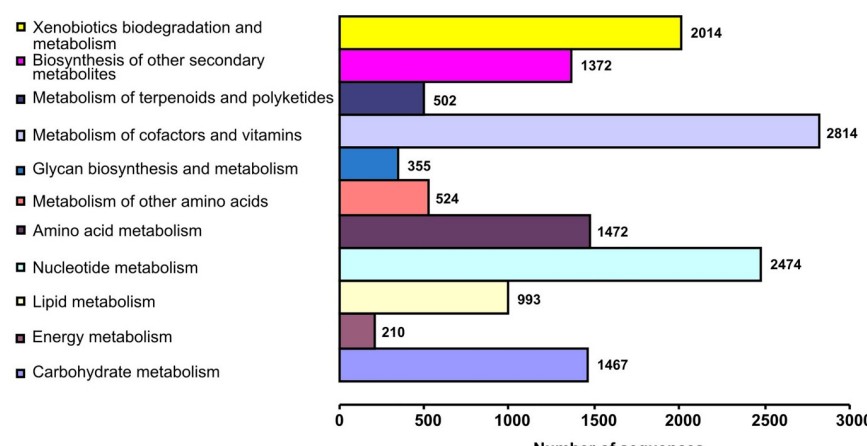

**Fig 7. Annotations of *S. alba* genes against KEGG database.** (A) KEGG distribution of genes into biological categories. (B) Classification of genes in KEGG metabolism category into sub-categories.

identified to produce seven enzymes that participated in cutin, suberin and wax biosynthesis pathway by KEGG analysis. These enzymes were O-acyltransferase (S-A_c_3385.g114), acyl-protein thioester reductase (S-A_c_8116.g84; S-A_c_318122.g9), seed peroxygenase (S-A_c_8126.g95; S-A_c_309255.g35), aldehyde oxygenase (deformylating) (S-A_c_25257.g84), O-feruloyl transferase (S-A_c_402.g243), O-fatty-acyltransferase (S-A_c_321813.g157; S-A_c_5996.g25) and fatty acyl-CoA reductase (S-A_c_10439.g247). All these genes were searched against the NCBI database by BLASTN to verify their roles in plant physical defense systems. The BLASTN results are listed in Table 2 and confirmed the KEGG database results for all these ten genes. The ECERIFERUM 1 (CER1) [44] like gene identified in *S. alba* genome encoded for fatty acid hydroxylase superfamily protein participated in aldehyde oxygenase (deformylating) activity. This is a key enzyme for biosynthesis of cuticular wax. The CER1 was also reported for increased pollen fertility in plants [45]. The phylogeny of *S. alba* CER1 like gene with homologous genes from Brassicaceae plants was illustrated in Fig 8 that they divided into two clades. The one set of *S. alba* CER1 like with *A. thaliana* CER1, *B. rapa* cv. chifu CER1, *Eutrema salsugenium* CER1, and *A. thaliana* CER3. Another set has *B. oleracea var oleracea*, *B. napus* cv ZS11 FAH1, *Capsella rubella* cv Monte FAH1, *Camelina sativa* cv DH55

**Table 2. List of predicted genes with BLASTN results against NCBI showing their role in plant physical defense.**

| S. No. | Predicted Gene from draft genome of S. alba | BLASTN (NCBI) species | Enzyme | E-value | Query cover (%) | Identity (%) |
|---|---|---|---|---|---|---|
| 1 | S-A_c_3385.g114 | *B. napus* | Acyltransferase like protein At3g26840 chloroplastic | 0 | 94 | 92.59 |
| 2 | S-A_c_8116.g84 | *B. napus* | Fatty acyl CoA reductase 2 | 0 | 78 | 92.72 |
| 3 | S-A_c_318122.g9 | *A. thaliana* FAR1 | Fatty acyl CoA reductase 1 | 0 | 92 | 86.10 |
| 4 | S-A_c_8126.g95 | *B. rapa* | Peroxygenase 4 | 0 | 100 | 95.68 |
| 5 | S-A_c_309255.g35 | *B. napus* | Peroxygenase 4 | 1e-170 | 84 | 96.49 |
| 6 | S-A_c_25257.g84 | *B. napus* Protein CER1 | Aldehyde decarbonylase | 0 | 100 | 97.25 |
| 7 | S-A_c_402.g243 | *B. oleracea var. oleracea* | Omega hydroxypalmitate O-feruloyl transferase | 0 | 100 | 94.33 |
| 8 | S-A_c_321813.g157 | *Raphanus sativus* | Long chain alcohol O-fatty acyl transferase 5 | 5e-115 | 97 | 90.91 |
| 9 | S-A_c_5996.g25 | *Arabis alpina* | O- fatty acyl transferase | 2e-43 | 82% | 77.78 |
| 10 | S-A_c_10439.g247 | *B. napus* | Fatty acyl CoA reductase 4 | 0 | 100 | 95.99 |

FAH1 and *A. lyrata* subsp *lyrata* FAH1. This study revealed that *S. alba* FAH superfamily gene was closely related to *A. thaliana* CER1 gene.

## 3.6. Estimation of retrotransposon abundance in *S. alba* genes

The retrotransposons played a well-known role in genome size expansion and evolution. They composed up to 50% mammalian genome and somewhat less in plant genomes. They are mobile elements and can transpose with RNA intermediate. Retrotransposons are present in high copy numbers in all types of plants such as angiosperms, gymnosperms, pteridophytes, etc. [46,47] The BLAST search and gene prediction criteria revealed that *S. alba* genome has a variety of retrotransposons. There were a total of 2293 genes in the genome that fall in the retrotransposon category. They were produced a total of about 2% part from all gene sequences identified in *S. alba* genome. The further classification of retrotransposon was found as Gag-Pol polyprotein/retrotransposon (999 genes), Copia-like (749 genes), Ta11-like non-LTR retrotransposon (236 genes), retrotransposon ORF-1 protein (125 genes), En/Spm-like transposon (82 genes), hAT transposon superfamily (52 genes) and retrotransposon Gag (48 genes). However, Gypsy-like retrotransposons were not identified in the genome.

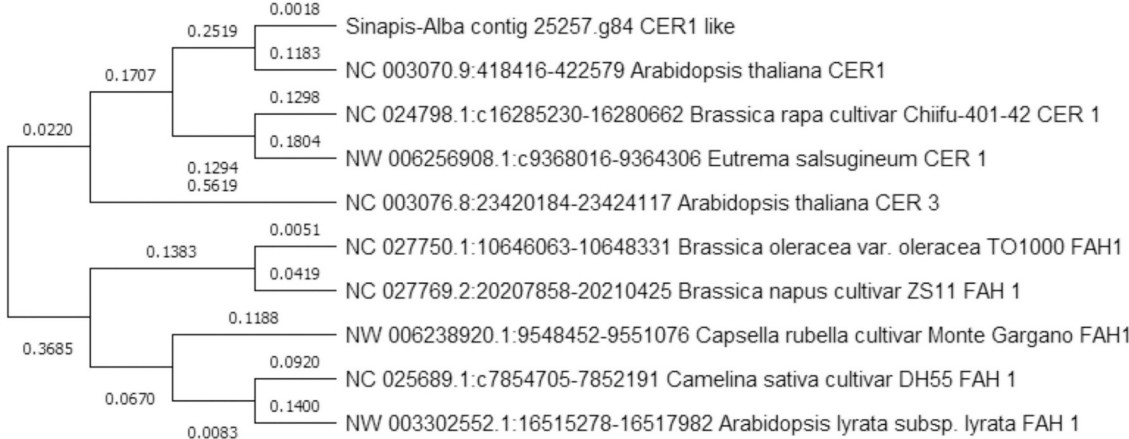

**Fig 8. UPGMA-based phylogeny of CER1 like gene identified in *S. alba* genome with other plant species of Brassicaceae.** The tree was constructed from gene sequence data obtained from NCBI and two clades were recognized.

## 4. Conclusions

The NGS sequencing data has become available in recent years because of cheaper cost and in reach of maximum workers to sequence large genomes. However, polyploid plants are still tough to sequence and develop reference sequence assembly. The Illumina is still a better choice for small cost sequencing and produces small reads. In the present study, we have developed the first draft genome assembly of *S. alba* from WGS sequences available in the SRA (NCBI) database. The assembled genome showed short contigs (403423) with an average size of 1138 bp and N50 was 1742 bp. The BLASTX search revealed a 32.9% sequence similarity with *A. thaliana*, a model plant of Brassicaceae family. The assembly produced a total of 70,731 unique SSRs with an average of one SSR per 5.88 kb genome. The validation experiment of randomly selected SSR markers has confirmed the amplification of more than 93 percent markers. The AUGUSTUS predicted 113107 genes and the majority (81.2%) of were less than 1000 base pairs with all IPScan IDs. Out of 113,107 predicted genes, 16119 genes have been identified to participate in various biochemical pathways to produce 1751 enzymes. During KEGG analysis, it was revealed that *S. alba* owns physical and biochemical defense-related pathways. The three most important anti-cancer pathways were identified during KEGG analysis related to humans that could be studied later. Besides, our approach was to provide information on assembly, functional annotation and SSR markers to exploit this plant for future research work in crop improvement, medicine (antibiotics and anti-cancer chemicals), resistance introgression and marker-assisted breeding programs.

## Supporting information

**S1 Table. Total in-silico SSR markers developed from *S. alba* genome assembly with complete details.**
(XLSX)

**S2 Table. Distribution of SSR markers in *S. alba* contigs with motifs, types, repetitions and positions.**
(XLS)

**S3 Table. Details of *S. alba* SSR markers used in the validation process.**
(XLS)

**S4 Table. Repeat masking report file of *S. alba* genome with information of masked bases.**
(PDF)

**S5 Table. KEGG pathways classifications file of *S. alba* genes.**
(DOC)

**S1 Fig.**
(TIF)

## Acknowledgments

We acknowledge NCBI for providing raw sequence data, and the Department of Energy, JGI, USA for allowing us to use this raw sequenced data.

## Author Contributions

**Conceptualization:** Preetesh Kumari.

**Data curation:** Preetesh Kumari.

**Funding acquisition:** Preetesh Kumari, Kaushal Pratap Singh.

**Investigation:** Preetesh Kumari, Kaushal Pratap Singh.

**Methodology:** Kaushal Pratap Singh.

**Resources:** Pramod Kumar Rai.

**Software:** Kaushal Pratap Singh, Pramod Kumar Rai.

**Supervision:** Pramod Kumar Rai.

**Validation:** Kaushal Pratap Singh.

**Visualization:** Preetesh Kumari.

**Writing – original draft:** Kaushal Pratap Singh.

**Writing – review & editing:** Preetesh Kumari, Kaushal Pratap Singh, Pramod Kumar Rai.

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
