## [Editor Report · Decision Letter 0]

15 Nov 2019

PONE-D-19-25413

Draft Genome of Multiple Resistance Donor Plant Sinapis alba: An insight into SSRs, Annotations and Phylogenetics

PLOS ONE

Dear Dr. Singh,

Thank you for submitting your manuscript to PLOS ONE. After careful consideration, we have decided that your manuscript does not meet our criteria for publication and must therefore be rejected.

In this work, the authors have used alredy published at 2006 (Zhang X et al., "De novo Transcriptome Analysis of Sinapis alba in Revealing the Glucosinolate and Phytochelatin Pathways", Front Plant Sci, 2016 Mar 4;7:259) publicly available Illumina NGS sequence data of S. alba genome (SRR3490913, SRR3490914, SRR1799170, SRR1799171) for de-novo assembly, gene prediction with annotation and identification of SSR markers.

The main result of the work is genes annotation, SSR detection, primer design and validation.

Unfortunately, the virtual detection of potential SSR loci in NGS data is not of practical importance. The authors needed to check all potential SSR pairs of primers on different Sinapis alba varieties or lines and publish only real and useful SSR pairs.

Authors wrote:

“For validation of randomly selected 51 SSR markers, the young leaves of field-grown S. alba plants were used for extraction and purification of genomic DNA as reported earlier [20].” However, there is no variety information or specific information.

I see that part of the SSR primers from supplemental table 3 are themselves part of the microsatellite sequence. Therefore, such primers cannot be used for SSR analysis:

cactctcactctctctctctctctc

ttcctctccctctccctctc

ttctctctctctctctctctcactc

ggcgcaaagagagagagaaa

cgtcatctctctctctcgcaat

accgacaaacaaactcgaca

gagacggagaaccagacgac

tgtgtgttggggatagatgg

taaggaagcaagaggcagga

tttcgaactccctctctctcc

I am sorry that we cannot be more positive on this occasion, but hope that you appreciate the reasons for this decision.

Yours sincerely,

Ruslan Kalendar, PhD

Academic Editor

PLOS ONE

- - - - -

---

## [Author Response · Author response to Decision Letter 0]

28 Nov 2019

I have responded on all points mentioned in decision letter by academic editor in rebuttal letter attached with manuscript file.

---

## [Editor Report · Decision Letter 1]

23 Jan 2020

PONE-D-19-25413R1

Draft Genome of Multiple Resistance Donor Plant Sinapis alba: An insight into SSRs, Annotations and Phylogenetics

PLOS ONE

Dear Dr. Singh,

Thank you for submitting your manuscript to PLOS ONE. After careful consideration, we feel that it has merit but does not fully meet PLOS ONE’s publication criteria as it currently stands. Therefore, we invite you to submit a revised version of the manuscript that addresses the points raised during the review process.

We would appreciate receiving your revised manuscript by Mar 07 2020 11:59PM. To enhance the reproducibility of your results, we recommend that if applicable you deposit your laboratory protocols in protocols.io, where a protocol can be assigned its own identifier (DOI) such that it can be cited independently in the future. For instructions see: http://journals.plos.org/plosone/s/submission-guidelines#loc-laboratory-protocols

We look forward to receiving your revised manuscript.

Kind regards,

Evangelia V. Avramidou, PhD

Sujan Mamidi, Ph.D.

Academic Editors

PLOS ONE

Journal Requirements:

4. Please include a copy of Tables 1 and 2 which you refer to in your text on line 159 and 285.

Additional Editor Comments (if provided):

The manuscript provides the annotation of whole genome of S. alba and the production of SSRs which will be useful for further analysis, due to the fact that S. alba present resistance in abiotic and biotic stress.

Some major point for the authors:

1. Did you get the sequences from NCBI portal or did you produce the sequences with the help of JGI as they write in assembly preparation (this part needs to be clear for the readers). I suppose that you produce the sequences but (as also other editor refers) there is no information of how was the DNA generated, leaf or root and the procedures. Also did you perform 4 different experiments or 4 different runs (this is also mentioned from the other editor's review).

2. This is not a whole genome research but contig generation (this is also mentioned from the other editor review)

3. Authors did only validation in one strain of S. alba, and they do not write how many plants of this strain have been used.

4. You have to specify the ploidy of the crop as also other editor mentioned and furthermore to do a synteny analysis because you found so many SSRs but without a previous genetic map further validation of the SSRs must be performed.

5. Furthermore whole Abstract should be rewritten especially in the part of background and in the conclusion section. Also abstract has more than 300 words (PlOs One requires that abstract should not exceed 300 words).

6. My concern is also according to the published article from Zhang X et al., 2006"De novo Transcriptome Analysis of Sinapis alba in Revealing the Glucosinolate and Phytochelatin Pathways", Front Plant Sci, 2016 Mar 4;7:259 authors also produced 14,727 SSRs and there is no discussion about advantages & disadvantages for use of SSR primers produced in current manuscript instead of using SSR primers from Zang et al 2016.I can understand that you used different material but I would expect better explanation about this point.

Finally, the manuscript needs further analysis and English editing before can be published in PlOS ONE.

---

## [Author Response · Author response to Decision Letter 1]

7 Mar 2020

The revised manuscript is submitting with all necessary changes suggested by academic editor and reviewers in their letter with thoroughly spelling corrections. The manuscript is formatted according to PLOS One journal requirement. I hope revised version will be suitable for publication.

---

## [Editor Report · Decision Letter 2]

16 Mar 2020

Draft Genome of Multiple Resistance Donor Plant Sinapis alba: An insight into SSRs, Annotations and Phylogenetics

PONE-D-19-25413R2

Dear Dr. Singh,

We are pleased to inform you that your manuscript has been judged scientifically suitable for publication and will be formally accepted for publication once it complies with all outstanding technical requirements.

With kind regards,

Evangelia V. Avramidou, PhD

Academic Editor

PLOS ONE

Additional Editor Comments (optional):

Dear authors,

the manuscript entitled "Draft Genome of Multiple Resistance Donor Plant Sinapis alba: An insight into SSRs, Annotations and Phylogenetics"

was significantly imroved after our suggestions and comments.

Therefore I thnik that it suitable for publication on the journal.

With kind regards
---

## [Editor Report · Acceptance letter]

23 Mar 2020

PONE-D-19-25413R2 

Draft Genome of Multiple Resistance Donor Plant *Sinapis alba*: An insight into SSRs, Annotations and Phylogenetics 

Dear Dr. Singh:

I am pleased to inform you that your manuscript has been deemed suitable for publication in PLOS ONE. Congratulations! Your manuscript is now with our production department. 

With kind regards,

on behalf of

Dr. Ruslan Kalendar 

Academic Editor

PLOS ONE